# Factors associated with endothelial cell density loss post Descemet membrane endothelial keratoplasty for bullous keratopathy in Asia

**Satoru Inoda**[1], **Takahiko Hayashi**[1,2,3]*, **Hidenori Takahashi**[1], **Itaru Oyakawa**[4,5], **Hideaki Yokogawa**[6], **Akira Kobayashi**[6], **Naoko Kato**[3,7], **Hidetoshi Kawashima**[6]

**1** Department of Ophthalmology, Jichi Medical University, Tochigi, Japan, **2** Department of Ophthalmology, University of Cologne, Cologne, Germany, **3** Department of Ophthalmology, Yokohama Minami Kyosai Hospital, Kanagawa, Japan, **4** Department of Ophthalmology, Heart Life Hospital, Okinawa, Japan, **5** Department of Ophthalmology, Ryukyu University, Okinawa, Japan, **6** Department of Ophthalmology, Graduate School of Medical Science, Kanazawa University, Kanazawa, Japan, **7** Minamiaoyama Eye Clinic, Tokyo, Japan

* takamed@gmail.com

**Data Availability Statement:** All relevant data are within the paper and its Supporting Information files.

## Abstract

### Purpose

To investigate the factors associated with endothelial survival after Descemet's membrane endothelial keratoplasty (DMEK) in eyes of Asian patients with bullous keratopathy (BK).

### Methods

In this retrospective, consecutive interventional case series, 72 eyes of 72 patients who underwent DMEK were evaluated. Best corrected visual acuity (BCVA) and corneal endothelial cell density (ECD) were assessed at 12 months postoperatively. Multiple regression analysis was performed to assess parameters such as age, sex, axial length, preoperative visual acuity, re-bubbling, the ratio of graft to cornea area, iris damage scores, types of filling gases, air or $SF_6$ volume in the anterior chamber (AC) on postoperative day 1, and ECD loss rates at 12 months postoperatively.

### Results

BCVA improved significantly at 12 months after DMEK ($P < .001$). The rate of ECD loss at 12 months after DMEK was 54.4 ± 16.1%. Multiple linear regression analysis showed that a larger ratio of graft to corneal area ($P = 0.0061$) and higher donor ECD ($P = 0.042$) were the primary factors for a lower ECD loss rate at 12 months after DMEK.

### Conclusion

A relatively larger graft size compared to the host cornea and more donor ECD might help endothelial survival in patients with BK. Moreover, for such patients, the surgeon should

**Funding:** Minamiaoyama Eye Clinic provided support in the form of salaries for author NK, and this work was supported by the Japan Eye Bank Association (http://www.j-eyebank.or.jp). The funders had no role in study design, data collection and analysis, decision to publish, or preparation of the manuscript.

**Competing interests:** The authors have declared that no competing interests exist.

attempt to use a relatively larger graft size when performing DMEK, particularly in Asian eyes.

## Introduction

Corneal endothelial keratoplasty (EK), i.e., replacing only the posterior layer of the cornea, has been established as the standard treatment for corneal endothelial dysfunction. Descemet stripping automated endothelial keratoplasty (DSAEK) and Descemet membrane endothelial keratoplasty (DMEK) are the two main methods for EK and are superior to full-thickness transplantation (penetrating keratoplasty; PKP) in terms of faster visual recovery and lower graft rejection rates [1]. DMEK replaces only a thin layer, including the Descemet's membrane, and is characterized by a markedly excellent visual outcome and low rejection rates [2]. Therefore, the number of DMEK procedures performed has increased in Western countries, particularly, in patients with Fuchs endothelial corneal dystrophy (FECD).

DMEK should be an ideal treatment for endothelial dysfunction; however, a potential drawback is the relatively large decrease in endothelial cell density (ECD) postoperatively, which is similar to PKP [1–3]. This decrease in ECD after corneal transplantation has been reported to be prominent in patients with a history of glaucoma surgery, graft rejection, and iris damage [4–8]. As an indication, FECD and bullous keratopathy (BK) account for approximately 49% and 17% of endothelial keratoplasty cases in the United States of America, respectively [9]. In the Netherlands, FECD and BK account for approximately 85% and for 10% of DMEK cases, respectively, and the number of EK procedures has been reported to be increasing [10,11]. In contrast, BK is the leading reason for endothelial keratoplasty in Asian countries, including Japan [12,13].

Although endothelial survival after DMEK has been reported to be worse in patients with BK than in those with FECD [10,14], this aspect has never been evaluated in detail, and the outcomes after DMEK in patients with BK warrant further evaluation. Herein, we investigated the clinical outcomes of DMEK and the factors that influence them in Asian patients with BK.

## Methods

### Study design

This retrospective, multi-center study was approved by the institutional review board of Jichi Medical University (JICHI 19–034) and adhered to the tenets of the Declaration of Helsinki. The study procedures followed all institutional guidelines and all patients provided informed consent before the procedures were performed.

### Analysis of clinical date

This retrospective multi-center study included 72 eyes of 72 patients with corneal endothelial dysfunction treated at Yokohama Minami Kyosai Hospital, Kanazawa University, and Hart Life Hospital in Japan, between January 2016 and March 2018. In patients with cataract, phacoemulsification, and intraocular lens (IOL) implantation, surgery was performed 1 month prior to DMEK. All patients underwent DMEK with either 100% air or 20% sulfur hexafluoride ($SF_6$) as the anterior chamber (AC) tamponade. After 2016, we switched from air to $SF_6$ gas. DMEK was performed by three different surgeons (H.T., I.O., and K.A.), as previously reported [12]. Stripping was performed in the area 0.25–0.5 mm larger than the graft size.

Most of the grafts were 7.75–8.25 mm. The inclusion criteria were those used for DMEK, and the follow-up period was > 12 months. The exclusion criteria were prior corneal surgery or FECD. All patients were of Asian (Japanese) ethnicity. There were no patients with anterior chamber IOL, and those who underwent pars plana vitrectomy and scleral fixation were included in this study. Those who underwent glaucoma surgeries were excluded.

All donors were obtained from Cornea Gen (https://corneagen.com/), preserved in Optisol (Chiron Ophthalmics, Irvine, California) and pre-stripped. The storage time was approximately 7 days, on account of the shipping time.

All patients attended the follow-up visits as per standard protocols. The evaluated parameters included the preoperative corneal diameter; graft size; postoperative (1, 3, 6, and 12 months) best corrected visual acuity (BCVA); central corneal thickness (CCT); corneal endothelial cell density (ECD); axial length (AXL); and gas volume in the AC on postoperative day 1.

Additional parameters assessed were the iris damage scores before DMEK, age, sex, preoperative visual acuity, re-bubbling, and ECD-loss rates at 12 months postoperatively. BCVA was measured as decimal visual acuity and converted to logarithm of the minimum angle of resolution (logMAR) units for statistical analysis.

The iris damage score was defined as laser iridotomy, iris depigmentation, or iris defects due to intraocular surgeries before DMEK, and were classified into 5 grades [8]. Briefly, grade 0 indicated no damage; grade 1, iris damage limited to a single quadrant; and grades 2, 3, and 4, notable damage in two, three, and four quadrants, respectively.

CCT was measured by anterior segment optical coherence tomography (AS-OCT) (SS1000, Tomey Corporation, Aichi, Japan) and evaluated by a corneal specialist (H.T). ECD was evaluated using a specular microscope (FA3509; Konan Medical Hyogo, Japan); AXL was measured by optical biometry (IOL Master 500, Carl Zeiss Meditec, Oberkochen, Germany).

## Graft to corneal area ratio

The mean horizontal and vertical diameters of the cornea (white-to-white) were recorded. The area of the cornea was obtained as the area of a circle using the mean value as the diameter. Previous reports on DSAEK showed a strong correlation between arc length parameters and corneal diameter [15]. Fig 1 shows the graft diameter (r) and host corneal diameter (R); the ratio of graft to cornea area was obtained as $(r^2/R^2)$.

## Statistical analyses

Statistical analyses were performed using JMP Pro software version 14.0.0 (SAS Institute, Cary, NC, USA). Associations between pre-existing characteristics and ECD loss rates 12

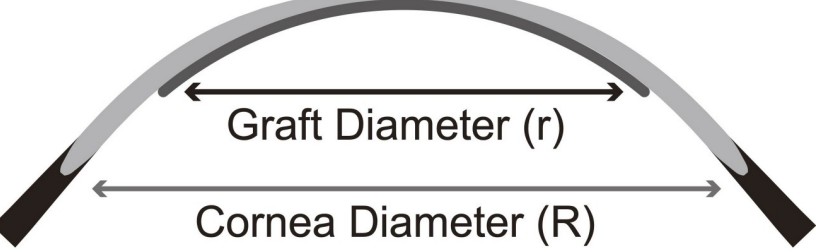

**Fig 1. The ratio of graft to cornea area.** The area ratio is the square of the ratio of the host corneal diameter to the graft diameter.

months after DMEK were examined using a regression line, multivariable regression analysis. Multivariate analysis was performed using ECD loss rates at 12 months after DMEK, age, AXL (mm), iris damage before DMEK, type of filling gases used, use of re-bubbling, preoperative BCVA, donor ECD, preoperative CCT, amount of the gases used in the AC, and the ratio of graft to cornea area as explanatory variables. Statistical significance was defined as $P < 0.05$.

## Results

### Patient characteristics

Table 1 summarizes the characteristics of patients who underwent DMEK in the current study. Most patients were older, and less than a third were male. Air or $SF_6$ was injected into the AC of 50 and 22 eyes, respectively. There was no statistically significant correlation between those who received air and those who received $SF_6$ in terms of postoperative BCVA, ECD loss rates at 12 months after DMEK, or re-bubbling rate ($P = 0.52$, $P = 0.29$, and $P = 0.85$, respectively).

No eyes showed signs of pupillary block, microbial infection, endothelial rejection, folding, or overlapping. Partial detachment of the graft, requiring re-bubbling into the AC, was observed in 9 eyes within 7 days of DMEK; the graft showed complete attachment immediately after re-bubbling in all eyes. There was no primary graft failure.

The BCVA improved significantly from $0.90 \pm 0.47$ preoperatively to $0.073 \pm 0.13$ at 12 months postoperatively ($P < 0.001$).

### Factors associated with ECD survival rates

The mean donor ECD and the ECD at 12 months after DMEK were $2715 \pm 231$ cells/mm$^2$ and $1246 \pm 478$ cells/mm$^2$, respectively. The mean ECD-loss rate at 12 months after DMEK was $54.4 \pm 16.1\%$. Univariate linear regression analysis revealed that graft size ($P = 0.0053$) and

**Table 1. Characteristics of patients.**

| Recipient characteristics | Eyes | |
|---|---|---|
| N | 72 | |
| Age (years), mean ± SD [range] | 74.5 ± 8.2 | [44–89] |
| Male, n (%) | 19 (26.4%) | |
| Preoperative BCVA (LogMAR), mean ± SD [range] | 0.90 ± 0.47 | [0.0458–2] |
| Postoperative BCVA (LogMAR), mean ± SD [range] | 0.073 ± 0.13 | [-0.0792–0.523] |
| Donor ECD (cells/mm$^2$), mean ± SD [range] | 2715 ± 231 | [2020–3313] |
| ECD 12 months after DMEK (cells/mm$^2$), mean ± SD [range] | 1246 ± 478 | [363–2519] |
| ECD-loss rates (%), mean ± SD [range] | 54.4 ± 16.1 | [15.1–86.1] |
| Axial length (mm), mean ± SD [range] | 23.2 ± 1.69 | [21–31.6] |
| Corneal diameter (mm), mean ± SD [range] | 11.0 ± 0.58 | [10–13] |
| Graft Size (mm), mean ± SD [range] | 7.87 ± 0.49 | [5–8.5] |
| Graft to corneal area ratio, mean ± SD [range] | 0.514 ± 0.067 | [0.207–0.625] |
| Air, n (%) | 50 (69.4%) | |
| Re-bubbling (+, [%]) | 9 (12.5%) | |
| Preoperative CCT (μm), mean ± SD [range] | 712 ± 94.0 | [501–956] |
| Iris damage scores before DMEK, mean ± SD [range] | 1.44 ± 0.82 | [0–4] |

BCVA: best corrected visual acuity, ECD: endothelial cell density, DMEK: Descemet membrane endothelial keratoplasty, CCT: central corneal thickness

**Table 2. Effect of the factors associated with ECD loss rate after DMEK from univariable analysis.**

| Regression analysis | Estimate | 95% CI | P-value |
|---|---|---|---|
| Age (years) | -0.0030 | [-0.0076, 0.0017] | 0.20 |
| Axial Length (mm) | 0.011 | [-0.015, 0.030] | 0.51 |
| Preoperative CCT (μm) | 0.00026 | [-0.00014, 0.00067] | 0.20 |
| Preoperative BCVA (LogMAR) | 0.054 | [-0.27, 0.13] | 0.18 |
| Donor ECD (cells/mm$^2$) | -0.00015 | [-0.00031, 0.000015] | 0.075 |
| Graft size (mm) | -0.11 | [-0.18, -0.033] | 0.0053 |
| Amount of Air in the AC | -0.18 | [-0.50, 0.13] | 0.24 |
| Iris damage | 0.046 | [-0.0000096, 0.091] | 0.051 |
| The ratio of graft to cornea area (%) | -0.0093 | [-0.015, -0.004] | 0.0009 |
| One-way ANOVA | Mean Squares | F ratio | P-value |
| Sex (female) | 0.018 | 0.69 | 0.41 |
| Air or Gas (Air) | 0.029 | 1.1 | 0.29 |
| Re-bubbling (+) | 0.0061 | 0.22 | 0.63 |

ECD: endothelial cell density, DMEK: Descemet membrane endothelial keratoplasty, CCT: central corneal thickness, BCVA: best corrected visual acuity, AC: anterior chamber

graft to corneal area ratio ($P < 0.001$) associated significantly with ECD-loss rates. (Table 2) Multiple linear regression analysis also identified the larger graft to corneal area ratio ($P = 0.0061$) and higher donor ECD ($P = 0.042$) to be associated with the lower ECD-loss rates (Table 3, Figs 2 and 3).

## Discussion

The current study showed the clinical outcomes after DMEK for BK in Asian eyes. Postoperative BCVA was significantly improved over preoperative BCVA without serious complications. Multivariable analysis showed that a relatively larger graft to host corneal area ratio and better donor ECD were importance factors for ECD survival after DMEK in Asian eyes.

For the outcome of ECD 12-months after DMEK, the ECD loss rates were 54.4 ± 16.1%, which were relatively high. Although many studies have reported excellent endothelial

**Table 3. Effect of the factors associated with ECD loss rate after DMEK from multiple liner regression analysis.**

| | Estimate | 95% CI | P-value | VIF |
|---|---|---|---|---|
| Age (years) | -0.00023 | [-0.0052, 0.0047] | 0.93 | 1.33 |
| Sex (female) | 0.032 | [-0.013, 0.076] | 0.16 | 1.25 |
| Axial Length (mm) | 0.014 | [-0.013, 0.040] | 0.30 | 1.60 |
| Preoperative CCT (μm) | 0.00013 | [-0.00035, 0.00060] | 0.60 | 1.59 |
| Preoperative BCVA (LogMAR) | -0.014 | [-0.11, 0.083] | 0.77 | 1.69 |
| Donor ECD (cells/mm$^2$) | -0.00017 | [-0.00034, -0.0000064] | 0.042 | 1.17 |
| Air or Gas (Air) | 0.011 | [-0.036, 0.058] | 0.63 | 1.53 |
| Amount of Air in the AC | -0.059 | [-0.38, 0.26] | 0.71 | 1.22 |
| Re-bubbling (+) | 0.022 | [-0.034, 0.077] | 0.44 | 1.11 |
| Iris damage | 0.026 | [-0.021, 0.074] | 0.28 | 1.22 |
| The ratio of graft to cornea area (%) | -0.00930 | [-0.0016, -0.0028] | 0.0061 | 1.51 |

ECD: endothelial cell density, DMEK: Descemet membrane endothelial keratoplasty, CCT: central corneal thickness, BCVA: best corrected visual acuity, AC: anterior chamber

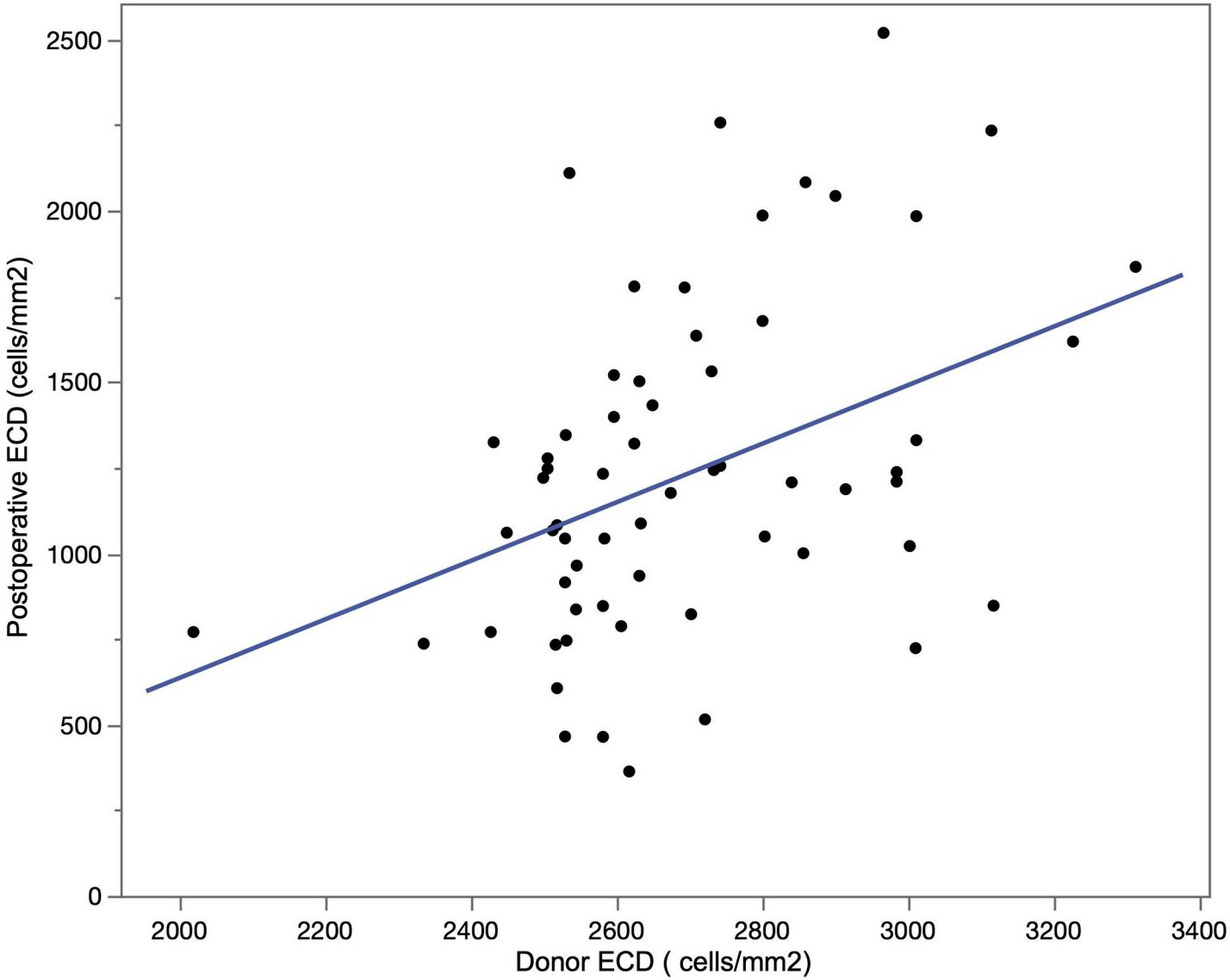

**Fig 2. ECD after 12 months vs. donor ECD.** Scatter plot showing that higher donor ECD is significantly associated with higher ECD after 12 months ($P < 0.001$).

outcomes after DMEK (ECD-loss rates of 20–40%), most assessed the outcome in patients with FECD; there have been few reports regarding the endothelial outcomes after DMEK for BK. The loss of ECD in eyes with BK in our study is consistent with that in a previous report from the Netherlands Institute for Innovative Ocular Surgery [14].

Previous studies have reported on the relationship between graft diameter and endothelial survival after DSAEK. Some reports have stated that a larger graft diameter and higher donor ECD were significantly associated with graft survival rates [8,16]. In contrast, Schrittenlocher et al. reported that postoperative ECD was not significantly associated with DMEK graft diameters in the range of 8–10 mm in FECD eyes. [17]. Although many studies have evaluated graft size, no study has evaluated the ratio of graft to cornea area. We considered that since the corneal diameter varies; it is essential to evaluate the ratio of graft to cornea area. Interestingly, the multivariable analyses showed that a relatively larger ratio of graft to corneal area and higher

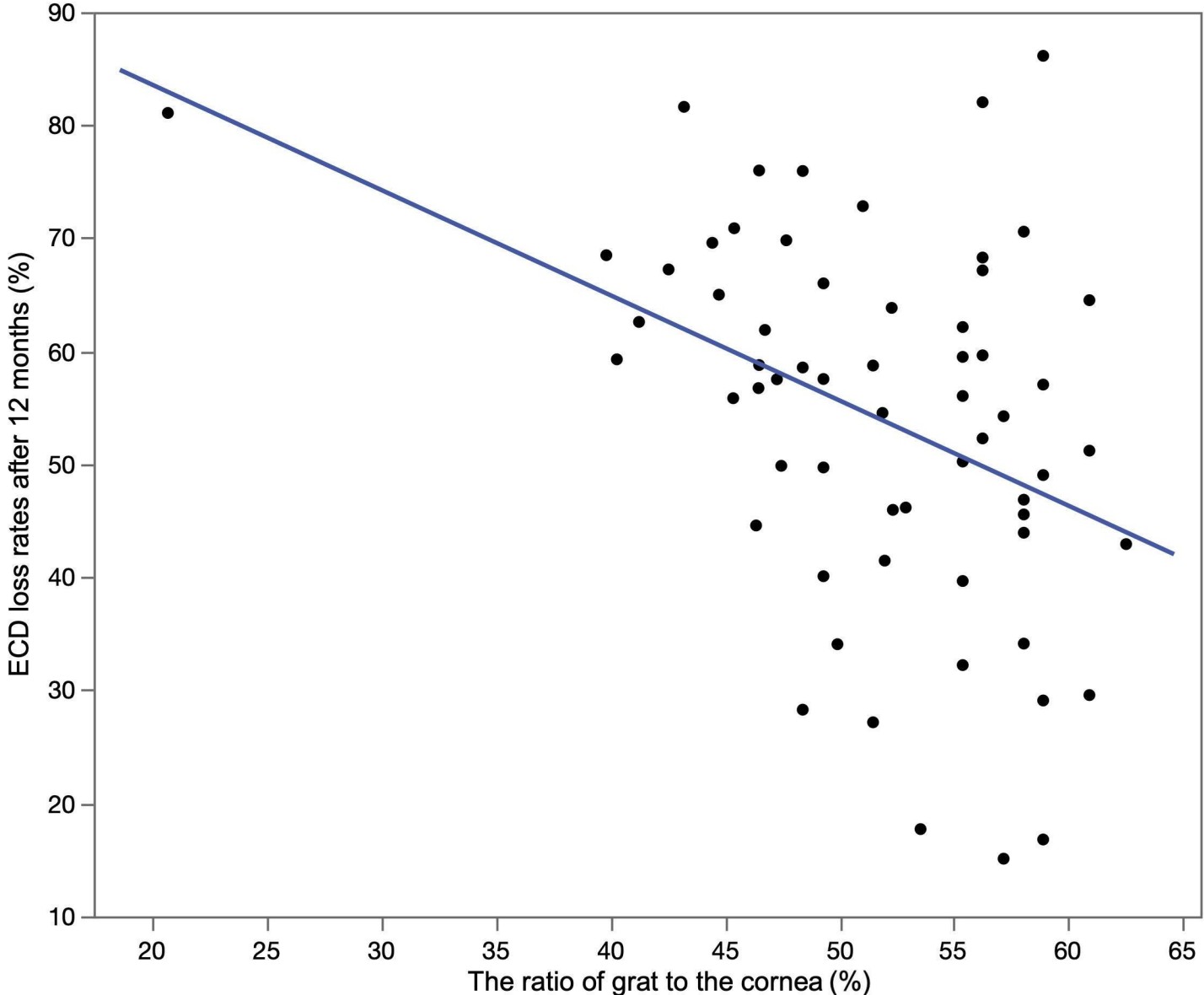

**Fig 3. ECD loss rates after 12 months vs. the ratio of graft to cornea area.** Scatter plot showing that the lager ratio of graft to the cornea is significantly associated with lower ECD loss rate after 12 months ($P < 0.001$).

donor ECD were important factors that determine ECD survival after DMEK for BK. There-fore, a larger graft diameter would possibly yield better endothelial outcome as theoretically, it might provide a higher ECD.

Although our results indicated that a larger graft size may be important for ECD survival, the question, "is larger better?" requires further consideration. When transplanting a larger graft, such as that in PKP, which includes more antigens, the incidence of graft rejection might be increased [18]. Using a larger graft may have a negative impact on re-bubbling rates because of overlap between the host Descemet's membrane and the graft. Furthermore, a larger graft requires more intricate surgical handling techniques, such as unfolding or insertion. Although our results indicated that a relatively larger ratio of graft to host cornea area was better for

ECD survival, there should be a plateau ratio, over which ECD survival would worsen. We attempted to evaluate this by drawing quadratic approximate expressions on 12-month ECD loss vs. the ratio of graft to cornea area scatter plots; however, we were unable to determine the significant ratio. Further research is needed to validate whether a larger graft is better for ECD survival or if there is an ideal graft size. In fact, cases with a larger graft diameter may make transplantation impossible.

Since a previous study reported a strong correlation between the anterior chamber depth and AXL [12], we used AXL in our analysis. Although physical damage to the ECD is more likely to occur during operation in a shallow AC [19], our analysis did not identify AXL as a potential factor of ECD loss. Moreover, Varadaraj et al. [20] compared ECD in eyes with open angles and those with untreated angle closure disease. They reported that in terms of primary closure suspects, eyes with a shallow AC had lower ECD than eyes with open angles. Furthermore, during the 12-month observation period of our study, the endothelial survival rate seemed to be relatively higher in eyes with short AXL, shallow AC. Further investigation regarding the association between AC/AXL and endothelial ECD after DMEK is required.

A strength of the current study is that our cohort did not include patients with FECD. Although patients with BK should have few healthy endothelial cells in the peripheral cornea, most patients with FECD might have some healthy endothelium in the peripheral cornea [21]. This difference of healthy endothelium distribution between BK and FECD may be one of the reasons why the results of the current study differed from those of Schrittenlocher et al. [17]. Although detailed knowledge regarding the migration of corneal endothelial cell is lacking, it is considered that corneal endothelial cells close the wound gap mainly via migration and increased cell spreading [22]. The creation of a wound induces the release of chemical signals that enhance cellular motility near the wound site; cells at the wound extend towards the free surface, producing a traction force directed towards the wound [23,24]. In FECD eyes with a peripheral rim of healthy endothelial cells, it has been shown that these healthy endothelial cells can be exploited to repopulate the centrally diseased endothelium either by migration or proliferation [25]. Further, a previous study suggested the possibility of treating corneal endo-thelial diseases by simple Descemet stripping without endothelial keratoplasty (DWEK) [26]. These studies suggest that healthy peripheral endothelium can migrate from the host cornea itself and be endothelialized. Another interesting study showed that adult endothelial cells were able to migrate in the human eye, supporting that endothelial cell in the donor graft can migrate toward the periphery of the host cornea [27,28]. The transplanted endothelium should compensate for the denuded area by migrating to the peripheral area (Fig 4) [28,29]. Thus, the decrease in ECD after DMEK could be attributed to the distribution of endothelial cells to the peripheral area, especially in BK eyes, although other factors could also play a role. In addition to the fact that the AC is shallow in the eyes of Asian individuals, this difference in distribution and the tendency for endothelial cell migration might cause relatively greater ECD loss in the present study as compared to that reported previously. Therefore, a relatively larger graft could be essential for ECD survival after DMEK for BK in Asian eyes.

Despite its limitations of being a retrospective study in a small number of patients, this study employed a precise and novel evaluation strategy using the larger ratio of graft to host cornea area. Although our sample size was small (72 eyes), no previous study has reported the outcomes of DMEK for BK in a larger sample to date.

In conclusion, the current study identified that a larger ratio of graft to cornea area is important for endothelium survival after DMEK in patients with BK. Moreover, higher donor ECD is important for endothelium survival. Thus, we suggest that surgeons should attempt to use relatively larger graft sizes when performing DMEK for patients with BK, particularly in Asian eyes.

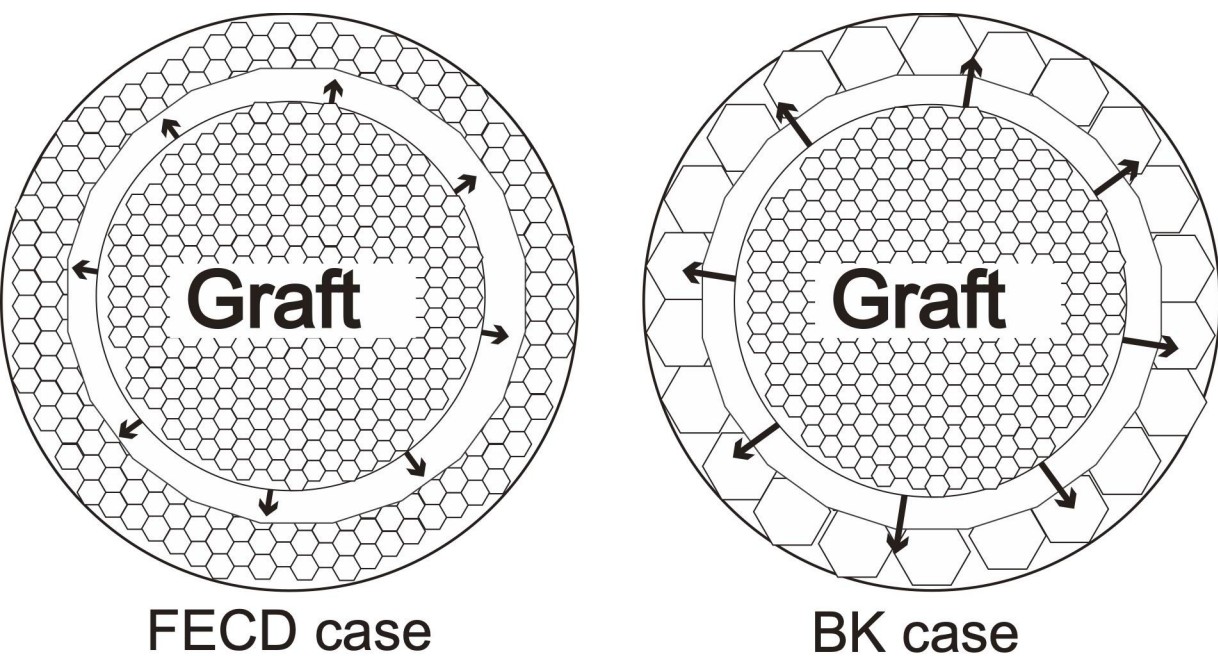

**Fig 4. The difference in the contribution of endothelial cell density (ECD) between Fuchs' endothelial corneal dystrophy (FECD) and bullous keratopathy (BK).** In BK, healthier endothelial cells supplied from the graft migrate to, and replace, the dysfunctional peripheral endothelium than in FECD.

## Supporting information

**S1 Dataset.**
(XLSX)

## Author Contributions

**Data curation:** Satoru Inoda, Takahiko Hayashi, Itaru Oyakawa, Hideaki Yokogawa, Akira Kobayashi, Naoko Kato.

**Formal analysis:** Takahiko Hayashi.

**Funding acquisition:** Takahiko Hayashi.

**Investigation:** Takahiko Hayashi, Itaru Oyakawa, Hideaki Yokogawa, Akira Kobayashi, Naoko Kato.

**Project administration:** Takahiko Hayashi, Hidenori Takahashi, Naoko Kato.

**Software:** Takahiko Hayashi, Hidenori Takahashi.

**Supervision:** Takahiko Hayashi, Akira Kobayashi, Hidetoshi Kawashima.

**Writing – original draft:** Satoru Inoda, Takahiko Hayashi.

**Writing – review & editing:** Takahiko Hayashi, Hidenori Takahashi, Akira Kobayashi, Naoko Kato.

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
