## [Decision Letter · Decision Letter 0]

31 Mar 2020

PONE-D-20-03729

Factors Associated with Endothelial Cell Density Loss Post Descemet Membrane Endothelial Keratoplasty for Bullous Keratopathy in Asia

PLOS ONE

Dear Dr. Hayashi,

Thank you for submitting your manuscript to PLOS ONE. After careful consideration, we feel that it has merit but does not fully meet PLOS ONE’s publication criteria as it currently stands. Therefore, we invite you to submit a revised version of the manuscript that addresses the points raised during the review process.

We would appreciate receiving your revised manuscript by May 15 2020 11:59PM. To enhance the reproducibility of your results, we recommend that if applicable you deposit your laboratory protocols in protocols.io, where a protocol can be assigned its own identifier (DOI) such that it can be cited independently in the future. For instructions see: http://journals.plos.org/plosone/s/submission-guidelines#loc-laboratory-protocols

We look forward to receiving your revised manuscript.

Kind regards,

Hidenaga Kobashi, M.D., Ph.D.

Academic Editor

PLOS ONE

Additional Editor Comments (if provided):

The reviewers and editor have completed their assessments of your manuscript, "Factors Associated with Endothelial Cell Density Loss Post Descemet Membrane Endothelial Keratoplasty for Bullous Keratopathy in Asia" (PONE-D-20-03729) and would like to publish it in the journal once you have responded to the referees' comments (enclosed below). In the cover letter with the revised manuscript, please indicate how each of the reviewers' suggestions was addressed.

Journal Requirements:

We note that one or more of the authors are employed by a commercial company: Minamiaoyama Eye Clinic

Reviewers' comments:

Reviewer's Responses to Questions

**Comments to the Author**

1. Is the manuscript technically sound, and do the data support the conclusions?

Reviewer #1: Yes

Reviewer #2: Yes

2. Has the statistical analysis been performed appropriately and rigorously? 

Reviewer #1: Yes

Reviewer #2: Yes

3. Have the authors made all data underlying the findings in their manuscript fully available?

Reviewer #1: No

Reviewer #2: Yes

4. Is the manuscript presented in an intelligible fashion and written in standard English?

Reviewer #1: Yes

Reviewer #2: Yes

5. Review Comments to the Author

Reviewer #1: This study evaluated parameters associated with 12-month endothelial cell loss in Japanese eyes treated with DMEK for bullous keratopathy.

1. Introduction, line 66: please use data from the more recent 2018 EBAA report instead of the 2015 report.

2. In Table 1, please report the minimum and maximum values for each parameter.

3. Please recheck all the citation numbers throughout the text, because many are wrong. For example, in line 201 the citation should probably be 12 instead of 15; in line 204 the citations should probably be 18,19 instead of 16,17; and in line 205 the citation should be 19 instead of 17.

4. Line 205 should note that the majority of patients in Schrittenlocher’s study were treated for FECD.

5. Line 259, please clarify what is meant by “may be better at least around 65%”.

6. Please add 2 scatter plots, one showing 12-month ECD vs. donor ECD and the other showing 12-month cell loss vs. the ratio of graft to cornea area. These will help the reader better assess the clinical significance of these parameters.

Reviewer #2: Interesting, well written and well conducted study on the risk factors for EC loss after DMEK for bullous keratopathy in Asian eyes. Not much is known about that yet.

Major comments

-Discussion: how do authors explain the relatively high EC loss rates after 12 months? Axial length seems to be within comparable rates? Please give AC-Depth in Table 1.

-Does AC depth correlate with EC loss?

-149: How do you explain absent beneficial effect of SF6 use on detachment rates in you cohort in contrast to what is published?

-259: how did you calculate the optimal ratio of 65%?

-Discussion: when comparing your data with the Schrittenocher paper: please also give the absolute diameters you used in mm.

-Figures: please better support your migration concept with ref. from the literature.

Minor comments

-Numerous spelling errors in the references need to be corrected. Some grammar and spelling errors in the text need correction.

6. PLOS authors have the option to publish the peer review history of their article (what does this mean?). If published, this will include your full peer review and any attached files.

Reviewer #1: No

Reviewer #2: No

---

## [Author Response · Author response to Decision Letter 0]

12 May 2020

Reviewer #1: This study evaluated parameters associated with 12-month endothelial cell loss in Japanese eyes treated with DMEK for bullous keratopathy.

1. Introduction, line 66: please use data from the more recent 2018 EBAA report instead of the 2015 report.

Thank you for pointing this out. We have rechecked the report from EBAA and other reports, and subsequently corrected the description as follows: (Line 65–70)

“As an indication, FECD and bullous keratopathy (BK) account for approximately 49% and 17% of endothelial keratoplasty cases in the United States of America, respectively [9]. In the Netherlands, FECD and BK account for approximately 85% and for 10% of DMEK cases, respectively, and the number of EK procedures has been reported to be increasing [10,11].” 

2. In Table 1, please report the minimum and maximum values for each parameter.

Thank you for pointing this out. We have added the minimum and maximum values for each parameter in Table 1. 

Recipient characteristics Eyes 

N 72 

Age (years), mean, ±SD, [range] 74.5 ± 8.2 [44 – 89]

Male, n (%) 19 (26.4%) 

Preoperative BCVA (LogMAR), mean ± SD [range] 0.90 ± 0.47 [0.0458 – 2]

Postoperative BCVA (LogMAR), mean ± SD [range] 0.073 ± 0.13 [-0.0792 – 0.523]

Donor ECD (/mm2), mean ± SD [range] 2715 ± 231 [2020 – 3313]

ECD 12 months after DMEK (/mm2), mean ± SD [range] 1246 ± 478 [363 – 2519]

ECD-loss rates (%), mean ± SD [range] 54.4 ± 16.1 [15.1 – 86.1]

Axial length (mm), mean ± SD [range] 23.2 ± 1.69 [21 - 31.6]

Corneal diameter (mm), mean ± SD [range] 11.0 ± 0.58 [10 – 13]

Graft Size (mm), mean ± SD [range] 7.87 ± 0.49 [5 - 8.5]

Graft to host cornea area ratio (%), mean ± SD [range] 51.36 ± 6.65 [20.66 – 62.52] 

Air, n (%) 50 (69.4%) 

Re-bubbling (+, [%]) 9 (12.5%) 

Preoperative CCT (µm), mean ±SD [range] 712 ± 94.0 [501 – 956]

Iris damage scores before DMEK, mean ±SD [range] 1.44 ± 0.82 [0 – 4]

3. Please recheck all the citation numbers throughout the text, because many are wrong. For example, in line 201 the citation should probably be 12 instead of 15; in line 204 the citations should probably be 18,19 instead of 16,17; and in line 205 the citation should be 19 instead of 17.

Thank you for pointing this out. We have rechecked all the citation numbers. 

4. Line 205 should note that the majority of patients in Schrittenlocher’s study were treated for FECD.

Thank you for pointing this out. We agree with your comment and have now added a description. (Line 211–213, 245–249) 

“In contrast, Schrittenlocher et al. reported that postoperative ECD was not significantly associated with DMEK graft diameters in the range of 8–10 mm in FECD eyes.”　

“Although patients with BK should have few functional endothelial cells in the peripheral cornea, most patients with FECD might have some healthy endothelium in the peripheral cornea [20]. This difference of healthy endothelium distribution between BK and FECD may be one of the reasons why the results of current study differed from those of Schrittenlocher et al.” 

5. Line 259, please clarify what is meant by “may be better at least around 65%”.

Thank you for this comment. We evaluated the relation between 12-month ECD loss vs. the graft to host corneal area ratio, excluding the case of the lowest ratio of graft to the host cornea area (ratio = 20.7%). This figure is a scatter plot of 12-month ECD loss vs. the graft to host corneal area ratio showing a quadratic approximate expression (blue line) and a liner approximate expression (red line). Though there was not a significant association with a quadratic coefficient (p=0.35), this figure suggests that ECD might survive better at least around 60% to 65% of the ratio, especially in the BK eyes. Therefore, we stated that around 65% might be better. We have discussed this concept again with the co-authors, and have changed the description as follows. (line 226–232)

“Although our results indicated that a relatively larger ratio of graft to host cornea area was better for ECD survival, there should be a plateau ratio, over which ECD survival would worsen. We attempted to evaluate this by drawing quadratic approximate expressions on 12-month ECD loss vs. the ratio of graft to cornea area scatter plots; however, we were unable to determine the significant ratio. Further research is needed to validate whether a larger graft is better for ECD survival or if there is an ideal graft size.”

6. Please add 2 scatter plots, one showing 12-month ECD vs. donor ECD and the other showing 12-month cell loss vs. the ratio of graft to cornea area. These will help the reader better assess the clinical significance of these parameters.

Thank you for this suggestion. We have added the two scatters plots. (Fig 2, Fig 3)

Reviewer #2: Interesting, well written and well conducted study on the risk factors for EC loss after DMEK for bullous keratopathy in Asian eyes. Not much is known about that yet.

Major comments

-Discussion: how do authors explain the relatively high EC loss rates after 12 months? Axial length seems to be within comparable rates? Please give AC-Depth in Table 1.

-Does AC depth correlate with EC loss?

Thank you for these comments. Our previous study* showed that ACD was strongly associated with AXL, and there was no correlation between ACD and EC loss. Although some patients had records of ACD as well as AXL, we did not record ACD in all patients. Since our data set did not include ACD, it is impossible to evaluate the correlation. It will be essential to evaluate the association in a larger number of patients in the future. 

* Shimizu T, Hayashi T, Yuda K, Takahashi H, Oyakawa I, Yamazaki K, et al. Short axial length and iris damage are associated with iris posterior synechiae after Descemet membrane endothelial keratoplasty in Asian eyes. Cornea 2018; 37:1355-1359. https://doi.org/10.1097/ico.0000000000001698

-149: How do you explain absent beneficial effect of SF6 use on detachment rates in you cohort in contrast to what is published?

Thank you for this question. In the current study, 9 cases needed re-bubbling. Six cases were injected air, 3 were SF6. Though there was no significant difference in the current study, we think this was because the number of eyes that required re-bubbling was small. 

-259: how did you calculate the optimal ratio of 65%?

Thank you for this question. We evaluated the relation between 12-month ECD loss vs. the graft to host corneal area ratio, excluding the case of the lowest ratio of graft to the host cornea area (ratio = 20.7%). This figure is a scatter plot of 12-month ECD loss vs. the graft to host corneal area ratio showing a quadratic approximate expression (blue line) and a liner approximate expression (red line). Though there was not a significant association with a quadratic coefficient (p=0.35), this figure suggests that ECD might survive better at least around 60% to 65% of the ratio, especially in the BK eyes. Therefore, we stated that around 65% might be better. We have discussed this concept again with the co-authors, and have changed the description as follows: (line 226–232)

“Although our results indicated that a relatively larger ratio of graft to host cornea area was better for ECD survival, there should be a plateau ratio, over which ECD survival would worsen. We attempted to evaluate this by drawing quadratic approximate expressions on 12-month ECD loss vs. the ratio of graft to cornea area scatter plots; however, we were unable to determine the significant ratio. Further research is needed to validate whether a larger graft is better for ECD survival or if there is an ideal graft size.”

-Discussion: when comparing your data with the Schrittenocher paper: please also give the absolute diameters you used in mm.

Thank you for this suggestion. We always determined the graft size depending on the host cornea. As shown in our data set, most of our grafts were 7.75–8.25 mm. We have added a description regarding the absolute diameter we used. (line 93–94)

“Most of the grafts were 7.75 – 8.25 mm.”

-Figures: please better support your migration concept with ref. from the literature.

Thank you for this suggestion. As you mention, there are two possibilities of migration: migration from the host cornea, and from the donor cornea. We have added the references, and revised the manuscript according to your suggestion. (Line 249–263; references 21–28) 

“Although detailed knowledge regarding the migration of corneal endothelial cell is lacking, it is considered that corneal endothelial cells close the wound gap mainly via migration and increased cell spreading [22]. The creation of a wound induces the release of chemical signals that enhance cellular motility near the wound site; cells at the wound extend towards the free surface, producing a traction force directed towards the wound [23,24]. In FECD eyes with a peripheral rim of healthy endothelial cells, it has been shown that these healthy endothelial cells can be exploited to repopulate the centrally diseased endothelium either by migration or proliferation [25]. Further, a previous study suggested the possibility of treating corneal endothelial diseases by simple Descemet stripping without endothelial keratoplasty (DWEK) [26]. These studies suggest that healthy peripheral endothelium can migrate from the host cornea itself and be endothelialized. Another interesting study showed that adult endothelial cells were able to migrate in the human eye, supporting that endothelial cell in the donor graft can migrate toward the periphery of the host cornea [27,28].”

Minor comments

-Numerous spelling errors in the references need to be corrected. Some grammar and spelling errors in the text need correction.

Thank you for pointing this out. We have rechecked all the citation numbers and corrected the grammatical/spelling errors throughout the manuscript.

---

## [Editor Report · Decision Letter 1]

21 May 2020

Factors Associated with Endothelial Cell Density Loss Post Descemet Membrane Endothelial Keratoplasty for Bullous Keratopathy in Asia

PONE-D-20-03729R1

Dear Dr. Hayashi,

We are pleased to inform you that your manuscript has been judged scientifically suitable for publication and will be formally accepted for publication once it complies with all outstanding technical requirements.

With kind regards,

Hidenaga Kobashi, M.D., Ph.D.

Academic Editor

PLOS ONE

Additional Editor Comments (optional):

The reviewers have completed their assessments in this revision.

I would recommend to publish in Plos One.
---

## [Editor Report · Acceptance letter]

1 Jun 2020

PONE-D-20-03729R1 

Factors Associated with Endothelial Cell Density Loss Post Descemet Membrane Endothelial Keratoplasty for Bullous Keratopathy in Asia 

Dear Dr. Hayashi:

I am pleased to inform you that your manuscript has been deemed suitable for publication in PLOS ONE. Congratulations! Your manuscript is now with our production department. 

With kind regards,

on behalf of

Dr. Hidenaga Kobashi 

Academic Editor

PLOS ONE